# Beneficial Effects of a Mixture of Algae and Extra Virgin Olive Oils on the Age-Induced Alterations of Rodent Skeletal Muscle: Role of HDAC-4

**DOI:** 10.3390/nu13010044

**Published:** 2020-12-25

**Authors:** Daniel González-Hedström, Teresa Priego, Asunción López-Calderón, Sara Amor, María de la Fuente-Fernández, Antonio Manuel Inarejos-García, Ángel Luis García-Villalón, Ana Isabel Martín, Miriam Granado

**Affiliations:** 1Department of Physiology, Faculty of Medicine, Autonomous University of Madrid, 28049 Madrid, Spain; dgonzalez@pharmactive.eu (D.G.-H.); sara.amor@uam.es (S.A.); maria.delafuente@uam.es (M.d.l.F.-F.); angeluis.villalon@uam.es (Á.L.G.-V.); miriam.granado@uam.es (M.G.); 2Pharmactive Biotech Products S.L. Parque Científico de Madrid, Avenida del Doctor Severo Ochoa, 37 Local 4J, 28108 Alcobendas, Spain; aminarejos@hotmail.com; 3Department of Physiology, Faculty of Medicine, Complutense University of Madrid, 28040 Madrid, Spain; alc@med.ucm.es (A.L.-C.); anabelmartin@med.ucm.es (A.I.M.); 4CIBER Fisiopatología de la Obesidad y Nutrición, Instituto de Salud Carlos III, 28029 Madrid, Spain

**Keywords:** aging, omega 3 fatty acids, extra virgin olive oil, sarcopenia, atrogenes, autophagy, HDAC-4

## Abstract

Aging is associated with a progressive decline in skeletal muscle mass, strength and function (sarcopenia). We have investigated whether a mixture of algae oil (25%) and extra virgin olive oil (75%) could exert beneficial effects on sarcopenia. Young (3 months) and old (24 months) male Wistar rats were treated with vehicle or with the oil mixture (OM) (2.5 mL/kg) for 21 days. Aging decreased gastrocnemius weight, total protein, and myosin heavy chain mRNA. Treatment with the OM prevented these effects. Concomitantly, OM administration decreased the inflammatory state in muscle; it prevented the increase of pro-inflammatory interleukin-6 (IL-6) and the decrease in anti-inflammatory interleukin-10 (IL-10) in aged rats. The OM was not able to prevent aging-induced alterations in either the insulin-like growth factor I/protein kinase B (IGF-I/Akt) pathway or in the increased expression of atrogenes in the gastrocnemius. However, the OM prevented decreased autophagy activity (ratio protein 1A/1B-light chain 3 (LC3b) II/I) induced by aging and increased expression of factors related with muscle senescence such as histone deacetylase 4 (HDAC-4), myogenin, and IGF-I binding protein 5 (IGFBP-5). These data suggest that the beneficial effects of the OM on muscle can be secondary to its anti-inflammatory effect and to the normalization of HDAC-4 and myogenin levels, making this treatment an alternative therapeutic tool for sarcopenia.

## 1. Introduction

Muscle mass decreases with age, leading to a condition called sarcopenia, and this disorder is related to reduced muscle function and strength, which finally leads to frailty [1]. The main factors involved in sarcopenia are loss of alpha motoneurons, increased inflammatory and oxidative mediators, and low levels of anabolic hormones [1]. Since sarcopenia is a very complex process, the impact of each factor in the development of this age-induced alteration is not clear.

Protein metabolism of the muscle is meticulously regulated by counteracted variations between protein synthesis and degradation, with this balance being compromised in aging muscle. On one hand, defects in protein trafficking, degradation, and removal such as alterations in autophagy, ubiquitination, and lysosomal degradation have been reported in advancing age [2,3,4]. On the other hand, hormone levels (growth hormone, insulin-like growth factor 1 (IGF-I), testosterone, and estradiol), which stimulate the mass and function of skeletal muscle, decrease with aging [1]. Among the main regulators of muscle growth and repair, IGF-1 plays a central role [5]. IGF-1 has been shown to induce muscle fiber hypertrophy, stimulate muscle differentiation and myogenesis [6], and also prevent sarcopenia [7]. Most of the circulating IGF-I comes from endocrine liver production or from autocrine/paracrine local synthesis, and its synthesis and secretion are regulated by different hormones, mainly pituitary GH, dietary factors, and different conditions (i.e., muscle overload and stretch) [8]. IGF-I acts predominantly via the IGF-I receptor (IGF-IR), but its action is regulated by six IGF-I binding proteins (IGFBPs), which can regulate the effect of IGF-I. IGFBP-3 and -5 are the main IGF-I binding proteins in skeletal muscle [9], but these binding proteins also exert IGF-I independent functions [10]. It is well known that circulating levels of IGF-I decrease in old animals [1]. Similarly, aging induces changes in muscle IGFBP-3 and -5 expressions, which may contribute to muscle wasting [11]. These alterations can be a consequence of the action of inflammatory factors, since aging is related with a low-grade but persistent inflammatory state. The age-induced increase of inflammatory cytokine expression such as interleukin 6 (IL-6) triggers proteolysis and the consequent loss of muscle mass [12] as well as metabolic alterations such as insulin resistance [13]. Sarcopenia is also caused by mitochondrial disorders including loss of muscle mitochondria, decreased expression of peroxisome proliferator-activated receptor gamma coactivator 1-alpha (PGC-1α), which is the main regulator of mitochondriogenesis, mitochondrial DNA mutations, and an increase in mitochondrial reactive oxygen species (ROS) emission [14].

Novel factors involved in sarcopenia have been recently elucidated. In this sense, histone deacetylase 4 (HDAC-4) is being proposed as a key component [15]. Its induction in skeletal muscle has been described in different models of muscle atrophy [16,17]. This enzyme is also upregulated in sarcopenia, both in experimental animals and in humans [18,19]. Furthermore, inhibition of this enzyme prevents denervation-induced loss of myosin heavy chain (MHC) isoforms and blocks the action of proteolytic factors such as muscle RING-finger protein-1 (MuRF1) [19]. The upregulation of HDAC-4 is accompanied by an increase in myogenin in different models of neurodegenerative-related muscle atrophy [20], and inactivation of HDAC4 suppresses muscle atrophy [17]. Thus, the HDAC-4-myogenin axis seems to have a central regulatory role in muscle remodeling. Therefore, HDACs are proposed as targets for new pharmaceutical tools to treat age-related alterations such as heart disease, diabetes, and sarcopenia [21].

To date, no pharmacological treatment for sarcopenia has shown significant efficacy, and the most effective intervention used focuses on physical exercise and resistance training [22]. Dietary interventions may be a safe and effective tool in the prevention/treatment of this condition. In fact, the beneficial effects of omega-3 long chain fatty acid (PUFA) supplementation have been greatly proven in different skeletal muscle alterations (reviewed in [23]). The two main active omega 3 PUFA are eicosapentenoic acid (EPA; 20:5*n* − 3) and docosahexenoic acid (DHA; 22:6*n* − 3), which exert beneficial global actions such as anti-inflammatory, antioxidant, and insulin sensitizing factors, and can also directly act on skeletal muscle [24]. Other dietary interventions such as extra virgin olive oil (EVOO) consumption can be also beneficial in aging-induced alterations [25]. Effects of EVOO on muscle mass and function are mainly attributed to its phenolic compounds, with important anti-inflammatory [26] and anti-oxidative properties [27]. Recent clinical trials have reported beneficial effects of the consumption of this oil in improving sarcopenia indicators [28].

In this study, we used a mixture of algae oil, rich in omega 3 PUFAs, and EVOO. This combination not only has the benefits of both types of oils, but also protects omega-3 PUFA from oxidation, thanks to the high antioxidant capacity of EVOO [29]. The main dietary source of DHA is fish and fish oil. A disadvantage of fish and fish oil supplements is contamination with toxins such as mercury and dioxins, all of them having harmful effects on health. In contrast, algae oil does not have detectable levels of these contaminants [30]. Thus, its use may be more beneficial. We have previously described that this mixture of oils is beneficial for attenuating cardio-metabolic alterations in aged rats [31]. Our hypothesis is that these oils may also exert positive effects on skeletal muscle. We focused our study on the main factors involved in muscle growth and repair such as components of the IGF-I axis as well as on inflammatory, atrophic, and autophagic factors. Furthermore, in order to elucidate the mechanisms involved in the actions of the oil mixture, we also analyzed novel factors that may play a key role in the development of sarcopenia such as the HDAC-4-myogenin axis.

## 2. Materials and Methods

### 2.1. Animals and Treatment

Young (3 months old) and old (24 months old) male Wistar rats were housed and maintained under standardized conditions of temperature (22–24 °C) and humidity (50–60%). All the animals were fed ad libitum with standard chow. The experiment followed the European Union Legislation guidelines for the care and use of laboratory animals. The Animal Care Committee of the Universidad Autónoma de Madrid and the Autonomic Government approved the animal procedure (PROEX 048/18).

Treatment consisted of a daily dose of 2.5 mL/kg of an oil mixture administered by gavage (intragastric tube) during 21-days to old animals (Old + Oils, *n* = 5). The oil mixture was composed of 75% EVOO (Cornicabra variety; 80% oleic acid and 60.0 mg/g of secoiridoids) (Aceites Toledo S.A., Los Yébenes, Spain) and 25% algae oil (*Schizochytrium* spp: 35% DHA, 20% EPA and 5% Docosapentaenoic (DPA), DSM, Heerlen, Netherlands). The fatty acid composition and the analysis of the phenolic fraction of the mixture have been previously described [29]. Old rats (control old, *n* = 8), together with the young animals (*n* = 11), received the placebo (2.5 mL/kg of tap water) following the same procedure as for treated animals.

Body weight gain (daily), and food and water intake (weekly) were controlled during the 21 days of treatment. After the experimental period, rats were fasted overnight. The animals were euthanized by injection of an overdose of sodium pentobarbital (100 mg/kg) and decapitated. Serum (from trunk blood) was collected and stored at −20 °C for IGF-1 assay. Left gastrocnemius muscles were removed, weighed, and stored at −80 °C for isolation of RNA and proteins.

### 2.2. Serum Insulin-Like Growth Factor-1(IGF-I) Measurements

Serum IGF-1 levels were analyzed by a Mouse/Rat IGF-1 Quantikine enzyme linked immunosorbent assay (ELISA) Kit from MP Biomedicals (Orangeburg, NY, USA), following the manufacturer’s protocols. Sensitivity and intra-assay variation were 8.4 pg/mL and 1.9–2.5%, respectively.

### 2.3. Quantitative Real-Time Polymerase Chain Reaction (RT-qPCR)

Total RNA was extracted from the gastrocnemius muscle (100 mg) by the Trisure (BIOLINE, London, UK) method and quantified with a BioPhotometer (Eppendorf International, Hamburg, Germany) spectrophotometer. Integrity of the RNA was confirmed by agarose gel electrophoresis.

One microgram of total RNA extracted from the gastrocnemius tissue was reversed transcribed into first strand complementary DNA (cDNA) using a Quantiscript Reverse Transcription Kit (Qiagen, Valencia, CA, USA).

Quantification of RNA expression by real-time PCR was performed from a diluted cDNA template (10 ng total RNA equivalents), forward and reverse specific primers (300 nM, from Roche Diagnostics, Madrid, Spain), and 1 × Takara SYBR Green Premix Ex Taq (Takara BIO Inc., Otsu, Japan). The thermal cycling protocol was performed in iQ5 equipment (Bio-Rad, Madrid, Spain) and consisted of an enzyme activation step (95 °C for 10 s) followed by 40 cycles with a three step protocol (95 °C for 15 s, 60 °C for 30 s, and 72 °C for 30 s). In order to verify the purity of the products, a melting curve was produced after each run according to the manufacturer’s instructions. The threshold cycle (Ct) was calculated by the instrument’s software (iQ5 Optical System Software, v2.0, Bio-Rad, Madrid, Spain) and results were expressed relative to the average Ct from young animals (where the relative mRNA abundance has been arbitrarily set to 1) using the 2^−ΔΔCt^ method; 18 s was used as reference gene.

### 2.4. Protein Quantification by Western Blot

Muscle tissues were homogenized in lysis buffer (Radioimmunoprecipitation assay (RIPA) buffer 10 μL/mg) with a protease inhibitor cocktail (phenylmethane sulfonyl fluoride 100 mM, sodium deoxycholate 12.5 mM sodium orthovanadate 12.5 mM, and with phosphatase inhibitors, all from Sigma-Aldrich, St. Louis, MO, USA). The homogenates were then centrifuged at 13,000 rpm for 30 min at 4 °C. The protein concentration of the supernatant was measured by a colorimetric assay (Bradford protein assay from Sigma-Aldrich, St. Louis, MO, USA). The protein extract (50 µg) was mixed (1:1) with Laemmli loading buffer (Bio-Rad, Madrid, Spain) and heated (95 °C) for 5 min. Afterward, it was resolved by electrophoresis on polyacrylamide 4%–20% gradient gels (Bio-Rad, Madrid, Spain) under reducing conditions. The resolved proteins were transferred onto nitrocellulose membranes and blocked with 5% non-fat dry milk and 0.1% Tween (Sigma-Aldrich, Madrid, Spain), in Tris-buffered saline. Ponceau-S staining (Bio-Rad, Madrid, Spain) was performed to ensure equal transfer prior to blocking. Membranes were probed overnight at 4 °C sequentially with antibodies against phospho-Akt (Ser473) (antibody ID: AB_2315049, 1:1000; Cell Signaling Technology; Danvers, MA, USA), Akt (antibody ID: AB_671714, 1:2000; Santa Cruz Biotechnology; Dallas, TX, USA), microtubule-associated protein-1 light chain 3 (LC3b) A/B (D3U4C) XP (antibody ID: 12741, 1:1000; Cell Signaling Technology; Danvers, MA, USA), and HDAC-4 (antibody ID: 7628, 1:2000; Cell Signaling Technology; Danvers, MA, USA). A stripping buffer (Restore Western Blot Stripping Buffer, Thermo Scientific; Rockford, IL, USA) was used before adding each new antibody to the membranes. After incubation during 90 min with the appropriate secondary antibody conjugated to horseradish peroxidase (anti-mouse immunoglobulin G (IgG)) (Amersham Biosciences; Little Chalfont, UK); anti-rabbit IgG (GE Healthcare; Chicago, IL, USA); anti-goat IgG (Santa Cruz Biotechnology; Dallas, TX, USA)), bands were visualized using peroxidase activity (enhanced chemiluminescent reagent from Amersham Biosciences, Little Chalfont, UK). Band intensities were quantified by densitometry using a PC-Image VGA24 (Thermo Scientific; Rockford, IL, USA) program for Windows.

### 2.5. Statistical Analysis

Data are expressed as means ± standard error of the mean (SEM), and differences among experimental groups were analyzed by one-way analysis of variance (ANOVA). Post-hoc comparisons were made by using subsequent least-significant difference (LSD) multiple range test. A *p* value of <0.05 was considered significant.

## 3. Results

### 3.1. Body Weight Increase, Gastrocnemius Muscle Weight, and Protein Levels

Figure 1A shows the body weight gain during the 21 days of treatment. While an increase in body weight was observed in young rats, there was a decrease in old rats (*p* < 0.01). Treatment with the oil mixture prevented, in part, the decrease in body weight (*p* < 0.05). The relative weight of gastrocnemius muscle of untreated old animals was significantly lower than that of young rats (Figure 1B, *p* < 0.01). Old treated rats showed higher gastrocnemius weight compared to untreated old animals (*p* < 0.05), although it remained significantly lower than in young animals (*p* < 0.01). In addition, aging also decreased total protein levels in the gastrocnemius muscle (Figure 1C, *p* < 0.01) and treatment with the oil mixture completely prevented this adverse effect (*p* < 0.01).

The expression of the main myosin heavy chains (MHC) expressed in the gastrocnemius, MHC-I, and IIa were decreased in old animals compared to those of young animals, where this decrease was more pronounced in type IIa fibers (Figure 1D, *p* < 0.05). The expression levels of these two MHC isoforms in treated animals showed no significant differences in comparison with the ones analyzed in young animals.

### 3.2. Expression of Cytokines and Metabolic Regulators on Gastrocnemius Muscle

Aging was associated with an increase in IL-6 and a decrease in IL-10 mRNA levels in gastrocnemius muscle (Figure 2A, *p* < 0.01 and *p* < 0.05, respectively), and treatment with the oil mixture completely prevented these alterations (*p* < 0.01 and *p* < 0.05, respectively). No differences were observed in the expression levels of tumor necrosis factor-α (TNF-α) and IL-1β, in either untreated or in the old treated animals.

The expression levels of peroxisome proliferator-activated receptor alpha (PPAR-α) (Figure 2B) showed no statistically significant differences in the old control rats compared to the expression of the young ones, but the mRNA levels of PGC-1α decreased (*p* < 0.05) in old rats. The oil mixture increased PPAR-α expression of old rats compared to the young ones (*p* < 0.01) and prevented the decrease of PGC-1α (*p* < 0.05).

### 3.3. The Insulin-Like Growth Factor I (IGF-I) System

Figure 3 shows the effects of aging on circulating IGF-I levels and mRNA expression of different components of the IGF-I system in the gastrocnemius muscle. As shown in Figure 3A, IGF-I circulating levels decreased with age (*p* < 0.01), and treatment with the oil mixture did not prevent this decrease. IGF-I mRNA expression in gastrocnemius muscle was not affected by aging. However, old rats treated with the oil mixture showed a lower level of IGF-I mRNA expression compared with both young and old control animals (*p* < 0.05). Expression of the IGF-I receptor mRNA (Figure 3B) increased in old animals, both in control and treated animals (*p* < 0.01 and *p* < 0.05, respectively). In contrast, the expression levels of the IGFBP-3 decreased with aging (*p* < 0.01), and treatment did not affect this decrease. Old rats showed increased expression levels of IGFBP-5 (*p* < 0.05), and treatment prevented this increase, whereas IGFBP-5 levels expressed by treated rats were similar to those of young animals.

Akt signaling was affected by aging, where the phospho-Akt/total Akt ratio was lower in old animals in comparison with the young ones (Figure 3C, *p* < 0.01). Treatment did not modify this decrease.

### 3.4. Expression of Atrogenes, Autophagic Regulators and LC3bII/I Ratio

Oil mixture treatment did not prevent the aging-induced increase of either of the atrogenes, Murf-1 and atrogin-1 mRNAs (Figure 4A, *p* < 0.01 for both).

Old animals expressed lower levels of B-cell lymphoma 2 (BCL2)/adenovirus E1B 19 kDa interacting protein (Bnip) in the gastrocnemius muscle than young rats (Figure 4B, *p* < 0.01), and treatment with the oil mixture did not prevent this decrease. The 1A/1B-light chain 3 (LC3b) mRNA expression levels did not change in any of the groups studied. However, the LC3b II/I ratio, an index of autophagy activity, significantly decreased in old animals (Figure 4C, *p* < 0.05), and the treatment totally prevented this decrease (*p* < 0.05).

### 3.5. The Histone Deacetylase 4 (HDAC-4)-Myogenin Axis

HDAC-4 and myogenin gene expression increased greatly in old animals (Figure 5A, *p* < 0.01). Treatment with the oil mixture prevented this increase, totally in the case of HDAC-4 expression levels (*p* < 0.01), and partially in the case of myogenin (*p* < 0.05). The same response pattern was observed with respect to the HDAC-4 protein levels (Figure 5B). Treatment with the oil mixture prevented the age-induced increase of this protein in the gastrocnemius muscle (*p* < 0.05).

## 4. Discussion

The results of this study show that treatment with an oil mixture composed of algae oil and EVOO can reduce age-induced impairments in skeletal muscle physiology. We focused the study on the gastrocnemius muscle because it is one of the most affected by aging [32]. Oil mixture administration improved the age-induced decrease in gastrocnemius mass, total protein concentration, and gene expression of MHC. These data are in accordance with those from both animal and human studies, which point out that omega-3 fatty acid intake is associated with enhanced rates of protein synthesis in skeletal muscle, suggesting that it may be useful to prevent sarcopenia [33]. In addition, recent studies have also described the usefulness of EVOO in sarcopenia prevention [28].

Most of the beneficial effects observed may be the result of the anti-inflammatory and anti-oxidant effects of the oil mixture [31]. Systemic low-grade inflammation is a characteristic of the aging process and has been involved in the progress of a number of aging-associated alterations such as diabetes, cardiovascular diseases, dementia, etc. [34]. We have previously described the higher profile of inflammatory cytokines and oxidative parameters in the same cohort of old animals [31]. In the present study, we observed that skeletal muscle in old rats had an increased level of pro-inflammatory cytokine IL-6 and a decreased level of anti-inflammatory cytokine IL-10. Development of sarcopenia has been previously associated with overexpression of pro-inflammatory factors, particularly IL-6, TNF-α, and C-reactive protein [35]. Local IL-6 infusion in skeletal muscle can directly induce its atrophy [36]. Treatment with the oil mixture was able to reduce circulating levels of IL-6 [31] and also its expression in skeletal muscle, leading to better muscle status. Omega-3 PUFA such as EPA and DHA are substrates for the production of anti-inflammatory and inflammation resolving mediators (i.e., resolvins), which inhibit pro-inflammatory gene transcription [37]. This could be the reason why the old animals treated with the oil mixture had the same inflammatory profile in serum [31] and in skeletal muscle as the young ones, as can be seen in the present study. This oil mixture also had anti-oxidant effects [31]; its administration to the same cohort of animals reduced the oxidative status in the liver, heart, and vasculature. Increased emission of reactive oxygen species (ROS) has proven to be involved in sarcopenia [38], which is another way that the oil treatment can reduce the harmful effects of these molecules. Concretely, the anti-oxidative effects may be due to the polyphenol components found in the mixture [29]. Other authors have reported the beneficial effects of polyphenols in reducing oxidative stress and improving aged-muscle performance [39].

In addition to the anti-inflammatory and anti-oxidant actions of the oil mixture, it also has potent anti-diabetogenic effects [31], which may also exert beneficial effects on skeletal muscle. Increasing the rate of fatty acid oxidation and mitochondrogenesis can help maintain glucose and fatty acid circulating levels and improve insulin sensitivity. Omega-3 PUFAs bind to the transcription factor activators PPARα and PGC-1α, triggering the expression of genes related with fatty acid utilization [40]. It has been reported that mitochondrial turnover is impaired in aged skeletal muscle [41], with reduced expression levels of PGC-1α as indicative of decreased mitochondrial biogenesis. The fact that oil mixture administration increased PPARα and PGC-1α expression in the gastrocnemius may be indicative of increased mitochondriogenesis. This is in accordance with the findings reported by Fiamoncini et al. [42] with fish oils. These authors observed that enhanced lipid metabolism contributed to the prevention of adiposity and glucose intolerance. Another way of action of these oils is through a direct effect of omega-3 fatty acids on muscle mass [23]. These fatty acids have been shown to increase anabolic signaling in skeletal muscle of both rodents and humans [43,44].

Oil mixture administration did not modify the decrease of circulating IGF-I levels, but it did prevent the increase in serum concentrations of insulin observed in old animals [31]. Thus, the improvement in muscle mass in the rats treated with oil does not seem to be secondary to circulating IGF-I changes, but rather to other circulating factors and/or to local regulation of growth and repair processes. Concomitantly with this decrease in circulating IGF-I levels, there was an increase in IGF-I receptor (IGF-IR) expression in the gastrocnemius muscle. The increased expression of IGF-IR may be the result of the negative feedback of IGF-I on its own receptor in the muscle [45]. In fact, there was a significant negative correlation between these two parameters (Pearson’s R= −0.542, *p* = 0.006, n = 24). Curiously, local expression of IGF-I decreased with the treatment, where this decrease was possibly related to the decreased circulating levels of IL-6 in old rats treated with the oil mixture [31]. This hypothesis is based on the fact that local IL-6 infusion in muscle increased IGF-I expression [36]. Authors concluded that the mechanism by which IL-6 induces muscle atrophy is by negatively affecting the IGF-I signaling pathway. Eventually, inhibition of downstream IGF-I signaling could induce an increase in local IGF-I production by a compensatory mechanism.

Our results show that IGFBP-3 gene expression decreased in aged rats, and treatment with the oil mixture did not prevent this decline. The age-induced decline of IGFBP-3 gene expression in skeletal muscle has been described by other authors [46], who also reported an increase in IGFBP-5 with age. Evidence indicates that IGFBP-5 has been related with the process of senescence [47] and plays a key role in apoptosis by interfering with the IGF signaling pathway [10]. Prevention of aging-induced overexpression with the oil mixture of IGFBP-5 may be a consequence, at least in part, of the decrease observed in circulating IL6 serum levels [31] and could contribute to the improvement of gastrocnemius muscle weight observed in the treated old animals. Similarly, different anti-inflammatory treatments blocked the increase of IGFBP-5 in animal models of arthritis [48,49], reducing muscle wasting.

HDAC-4 is a class IIa deacetylase with little ability to act on nuclear histones but with enough ability to repress transcription by binding to transcription factors and even acting directly on MHC and PGC-1α proteins [19]. This deacetylase is upregulated in skeletal muscle in different atrophic conditions [16] including sarcopenia [18] and may have an important role in regulating atrophy, since its inhibition prevents the loss of MHC isoforms and blocks the action of MuRF1 in denervation-induced atrophy [19]. In our model, age-induced increase in HDAC-4 mRNA and protein levels is prevented by treatment with the oil mixture. These results are in agreement with those described by other authors in vitro in macrophages, where treatment with DHA, but not with other fatty acids, reduced the expression levels of HDAC-4 [50]. Taking into account that HDAC-4 represses the expression of Dach2, a negative regulator of myogenin [51], the decrease in myogenin in old rats treated with the oil mixture may be the result of decreased HADC-4 expression. Myogenin, in addition to its role in myogenesis, has a key role in muscle atrophy, since it upregulates atrogenes. In this sense, mice lacking myogenin are resistant to neurogenic atrophy [52], denoting that this factor is essential in promoting the mechanisms leading to muscle wasting. In fact, the HDAC-4-myogenenin axis has been proposed as a marker of muscle atrophy [20]. In different models of muscle atrophy such as spinal cord atrophy, amyotrophic lateral sclerosis, and Huntington disease, upregulation of the HDAC-4-myogenin axis has been reported [20,53,54], highlighting the central role of this axis on muscle degeneration. Despite the improved muscle status with the oil treatment, the increase in atrogenes, MuRF1, and atrogin-1 were not prevented by the oil mixture. It has recently been reported that HDAC-5 inhibition is able to prevent disuse-induced upregulation of atrogin-1 and myogenin, whereas HDAC-4 inhibition was only able to prevent increased myogenin expression [55]. In addition, it has been proposed that HDAC-4 deacetylates MHC proteins, allowing MuRF-1 to ubiquitinate them, triggering their proteolysis [19]. The increase in HDAC4 can also be related to the age-induced decrease in PGC-1α mRNA levels. It has been reported that blockage of HDAC4 was able to prevent the loss of PGC-1α induced by denervation of skeletal muscle [19].

A decline in autophagic function is another characteristic of aging muscle [56], which has been proposed as the main cause of the aging process [57] due to impairment of autophagy, which damages structures and organelles such as mitochondria. In our study, the ratio between LC3b II and I, which is indicative of ongoing autophagy, was decreased in the old animals. Other authors have also reported this decrease in the gastrocnemius muscle of old animals [58], and showed that voluntary wheel running partially restored this effect. Similarly, treatment with the oil mixture prevented the age-induced reduction in autophagy, which is in line with the results of Shin et al., who reported that DHA supplementation in vitro was able to increase the levels of different autophagy markers including LC3b-II [59]. HDAC-4 may also be involved in this effect, since it has been reported that HDAC-4 modulates autophagy at the vascular level [60]. Thus, it can also have an important role in skeletal muscle, although further research is required to demonstrate this hypothesis.

HDAC4 may have a central role in sarcopenia (Figure 6), increasing muscle senescence and local inflammation. In this sense, when HDAC-4 expression is blocked, the levels of pro-inflammatory markers decrease [60]. The oil mixture prevents the age-induced increase of HDAC-4 expression; thus, the mechanism involved in this effect may be a consequence of the anti-inflammatory effect of the oils, as has been previously discussed. However, some components of these oils may act by directly interfering with the methylation of the HDAC-4 gene. As previously reported by Tremblay et al. [61], omega 3 fatty acid supplementation changes the methylation profile of some genes including HDAC-4. In fact, HDAC-4 has been proposed as a potential therapeutic target in different diseases [20,60]. However, to our knowledge, this is the first study that has shown the beneficial effects of these oils on the HDAC-4-myogenin axis in skeletal muscle. 

## 5. Conclusions

In summary, the administration of a mixture of algae oil, rich in omega-3 fatty acids, and EVOO oil to old animals reduced age-induced skeletal muscle wasting. These effects were accompanied by lower pro-inflammatory status and IGFBP-5 expression levels together with the upregulation of factors related with fatty acid oxidation, mitochondriogenesis, and autophagy. It is proposed that the HDAC-4-myogenein axis may play a key role in these effects, although more research must be done to demonstrate this assertion. A limitation of the present study is that the effect of treatment with the oil mixture on aging-induced decrease in muscle strength was not analyzed. Further functional studies are required to clarify the potential effect of the oil mixture on muscle strength.

## Figures and Tables

**Figure 1 nutrients-13-00044-f001:**
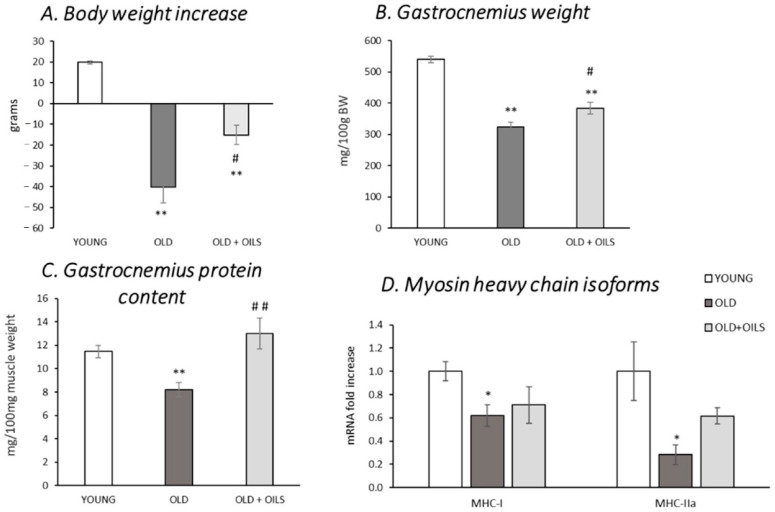
Beneficial effects of oil mixture on body weight gain (**A**), gastrocnemius muscle relative weight (**B**), gastrocnemius muscle protein concentration (**C**), and gastrocnemius mRNA levels of myosin heavy chain isoforms (MHC) I and IIa (**D**). Values are represented as mean ± standard error of the mean (SEM) of young rats (*n* = 11), old rats (*n* = 8), and old rats treated for 21 days with the oil mixture (OLD + OILS, *n* = 5). Statistics: ** *p* < 0.01 and * *p* < 0.05 vs. Young; ^##^
*p* < 0.01 and ^#^
*p* < 0.05 vs. Old, by least-significant difference post hoc analysis after significant one-way analysis of variance (ANOVA).

**Figure 2 nutrients-13-00044-f002:**
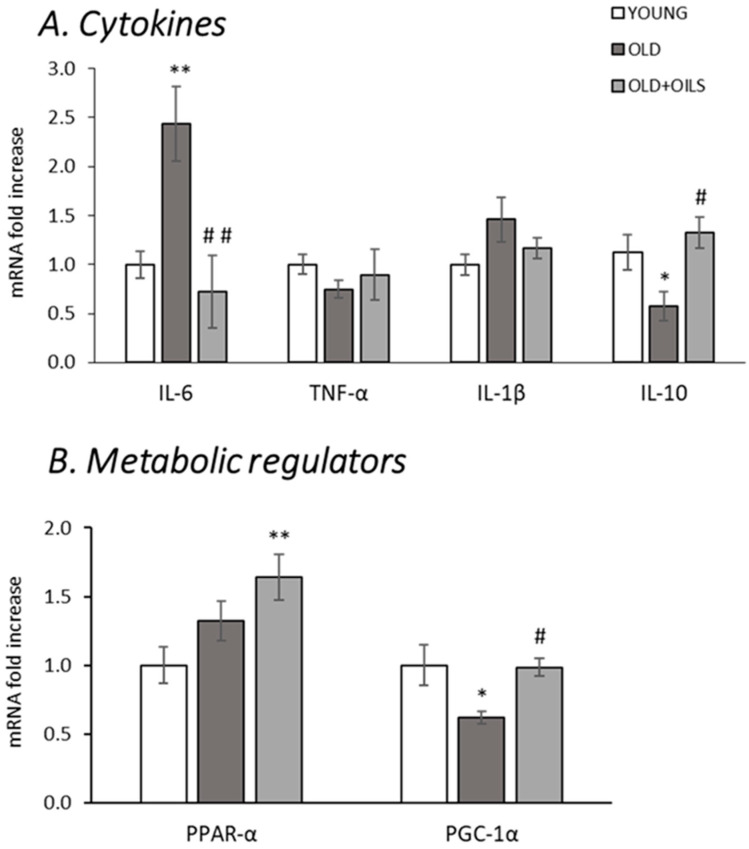
Oil mixture treatment improved the inflammatory profile and the expression of metabolic regulators in old gastrocnemius muscle. Figures show the mRNA levels of (**A**) interleukin-6 (IL-6), tumor necrosis factor alpha (TNF-α), IL-1β and interleukin-6 (IL-10), (**B**) peroxisome proliferator-activated receptor alpha (PPAR-α), and peroxisome proliferator-activated receptor gamma coactivator 1-alpha (PGC-1α) of young rats (*n* = 11), old rats (*n* = 8), and old rats treated for 21 days with the oil mixture (OLD + OILS, *n* = 5). Values are represented as mean ± standard error of the mean (SEM). Statistics: ** *p* < 0.01 and * *p* < 0.05 vs. Young; ^##^
*p* < 0.01 and ^#^
*p* < 0.05 vs. Old, by least-significant difference post hoc analysis after significant one-way analysis of variance (ANOVA).

**Figure 3 nutrients-13-00044-f003:**
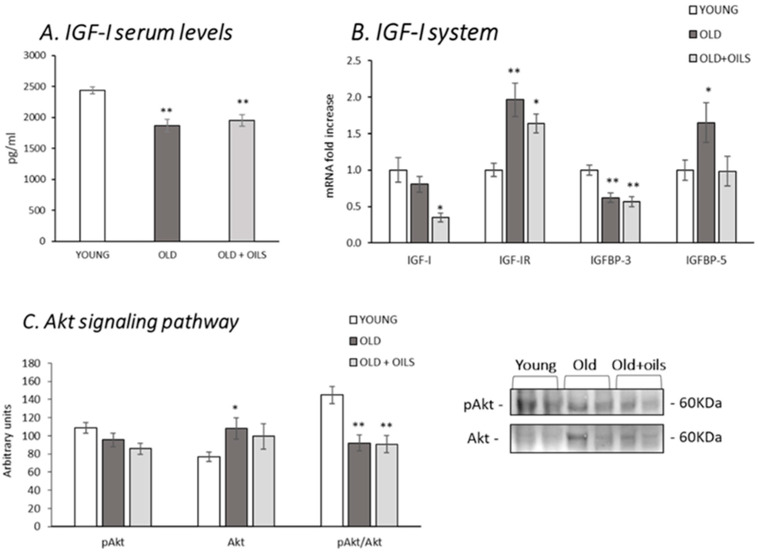
Oil mixture treatment did not prevent the age-induced decrease of IGF-I serum levels and the muscle Akt signaling pathway, although it did prevent increased expression of muscle IGFBP-5. Figures show (**A**) insulin-like growth factor (IGF-I) serum levels; (**B**) gastrocnemius IGF-I, IGF-I receptor (IGF-IR), IGF-I binding protein 3 (IGFBP-3), and 5 (IGFBP-5) mRNA levels; (**C**) gastrocnemius protein levels and ratio between phospho Akt (pAkt) and Akt of young rats (*n* = 11), old rats (*n* = 8) and old rats treated for 21 days with the oil mixture (OLD + OILS, *n* = 5). Values are represented as mean ± standard error of the mean (SEM). Statistics: ** *p* < 0.01 and * *p* < 0.05 vs. Young, by least-significant difference post hoc analysis after significant one-way analysis of variance (ANOVA).

**Figure 4 nutrients-13-00044-f004:**
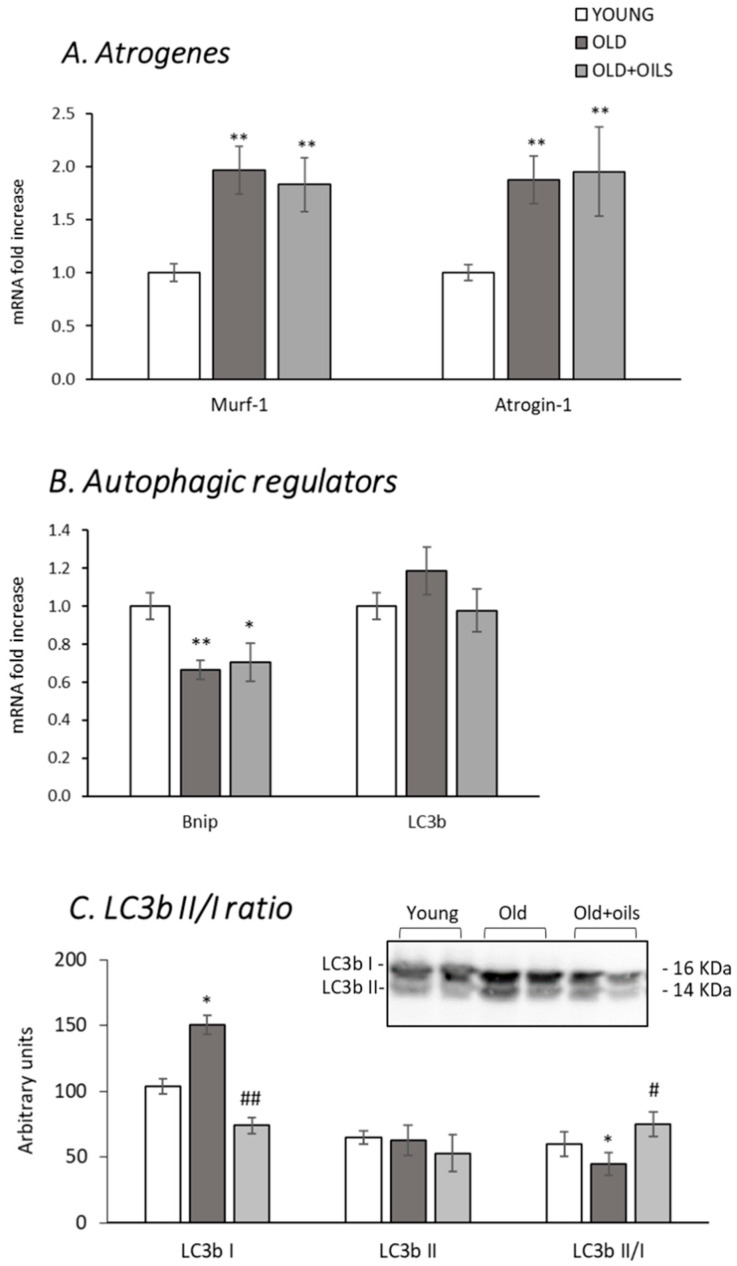
Oil mixture treatment did not prevent age-induced alterations in the expression of atrogenes and Bnip, although it did prevent a decrease in autophagy activity (ratio LC3b II/I) in skeletal muscle. Figures show gastrocnemius (**A**) muscle RING-finger protein-1 (MuRF1) and atrogin-1; (**B**) BCL2/adenovirus E1B 19 kDa interacting protein (Bnip) and 1A/1B-light chain 3 (LC3b) mRNA levels; (**C**) gastrocnemius protein levels and ratio between LC3b I and II of young rats (*n* = 11), old rats (*n* = 8) and old rats treated for 21 days with the oil mixture (OLD + OILS, *n* = 5). Values are represented as mean ± standard error of the mean (SEM). Statistics: ** *p* < 0.01 and * *p* < 0.05 vs. Young; ^##^
*p* < 0.01 and ^#^
*p* < 0.05 vs. Old, by least-significant difference post hoc analysis after significant one-way analysis of variance (ANOVA).

**Figure 5 nutrients-13-00044-f005:**
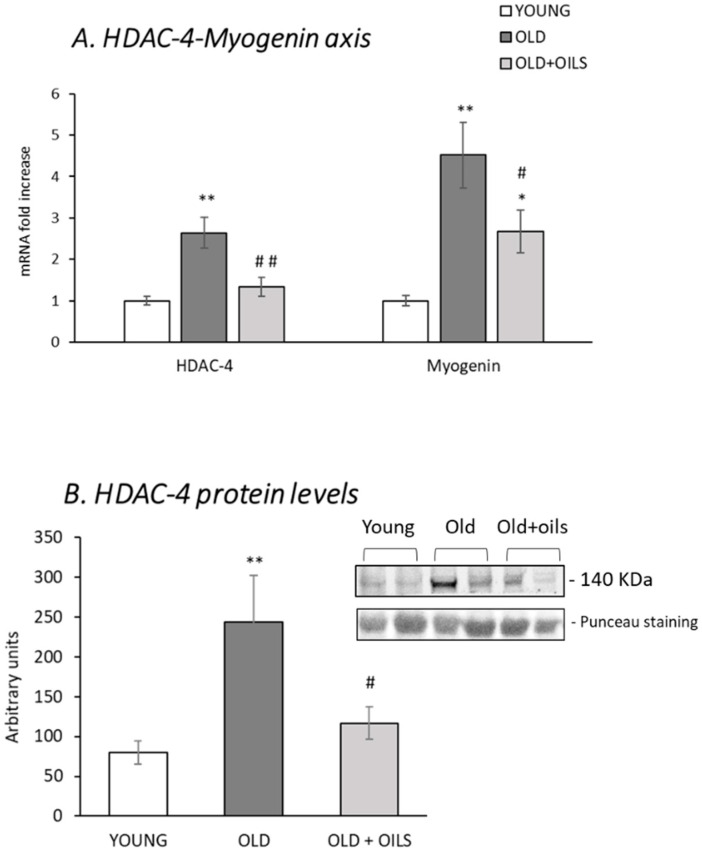
Oil mixture treatment prevented the age-induced activation of the HADC-4-myogenin axis in skeletal muscle. Figures show gastrocnemius mRNA levels of (**A**) histone deacetylase 4 (HDAC-4) and myogenin and (**B**) HDAC-4 protein levels of young rats (*n* = 11), old rats (*n* = 8) and old rats treated for 21 days with the oil mixture (OLD + OILS, *n* = 5). Values are represented as mean ± standard error of the mean (SEM). Statistics: ** *p* < 0.01 and * *p*< 0.05 vs. Young; ^##^
*p* < 0.01 and ^#^
*p* < 0.05 vs. Old, by least-significant difference post hoc analysis after significant one-way analysis of variance (ANOVA).

**Figure 6 nutrients-13-00044-f006:**
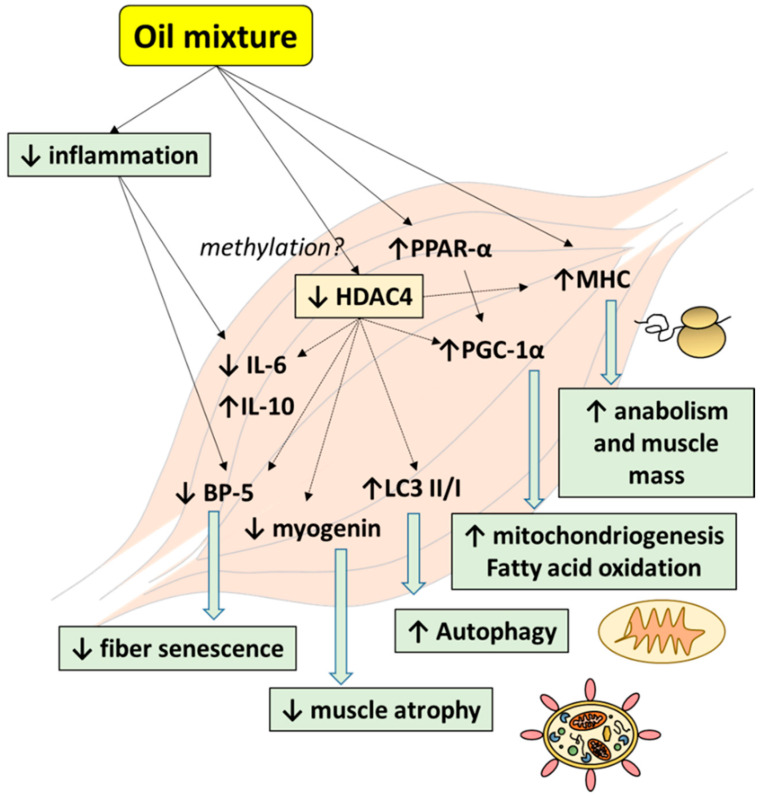
Schematic representation of the effects of the oil mixture on the main aged-induced alterations in skeletal muscle. Oil mixture administration prevented the age-induced increase of HDAC-4. This effect, together with the decrease in inflammatory mediators, may have beneficial consequences such as the prevention of the fiber senescence and muscle atrophy, restoration of autophagy equilibrium, increase in mitochondriogenesis, fatty acid oxidation rate, and protein synthesis. All these effects help to prevent the development of sarcopenia (see Discussion for further details). Abbreviations: BP-5 (insulin-like growth factor I binding protein 5); HDAC-4 (histone deacetylase 4); IL (interleukin); LC3 (1A/1B-light chain 3); MHC (myosin heavy chain); PGC-1α (peroxisome proliferator-activated receptor gamma coactivator 1-alpha); PPAR-α (proliferator-activated receptor alpha).

## Data Availability

Data available on request due to privacy restrictions.

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
