# Peer review of "Beneficial Effects of a Mixture of Algae and Extra Virgin Olive Oils on the Age-Induced Alterations of Rodent Skeletal Muscle: Role of HDAC-4"

_nutrients, 2020, doi:10.3390/nu13010044_

Round 1

Reviewer 1 Report

Summary

This article describes the potential effects of a mixture with algae and olive oil on sarcopenia in old rat model. The results show positive effects of the mixture on muscle by prevention of aging effects on the gastrocnemius. Mixture has also anti-inflammatory effects and seems to improve muscle metabolism, by a decrease of HDAC-4. Despite the effects on muscle protection, the IFG-I axis doesn’t seem implicated in muscle mass improvement.

Major issues

In this article the scientific question is valid and the sample size is correct. The methods used by the authors are appropriate in order to answer the question.

In this article, sarcopenia is defined by “progressive decline in skeletal muscle mass, strength and function”, but there is no test to evaluate the loss of strength, so how can the authors be sure that animals developed sarcopenia? Loss of muscle mass is not systematically associated with the loss of strength.

They did not mention the limitations of their study, it could be wise to mention it. Is there any adverse effects of the mixture oil for the rats?

Globally the language is clear and understandable, and the references are sufficient.

As mentioned in the minor issues, it should be important to precise the part of the gastrocnemius that the authors used. Moreover, it should be interesting to measure the same parameters in a second skeletal muscle with different muscle phenotype (ex. Soleus muscle).

The discussion has to be improved following several minor issues.

Minor issues

Introduction

The importance and the effects of IGF-1 impact on myogenesis and sarcopenia could be more developed, for a better understanding of IGF-1 part in the discussion. The scientific justification of algae oil utilization is poorly describing in the introduction.

Experimental materials

The authors can advise the number of animals used for each measure in the figure legend, and in the materials and methods. The authors written “Old male wistar (n=13)” then “half of the old rats”, so they have to precise the exact number.

Which part of the gastrocnemius was used? Because le superficial gastrocnemius is glycolytic and the deep part is more oxidative. The type of fiber is also different in function of the part of this muscle.

Why the authors did not use several muscles in order to validate the decrease of muscle mass in some other hind limb muscles?

Experimental results

It could be relevant to give more precisions in the figure legend, for example with an informative title that resume the major result of the figure. They should mentioned the samples numbers for each graphs.

The authors should measure all the MHC in gastrocnemius and should give the results for the major MHC present in Gastrocnemius: the MHC-IIb/x.

Because the results are not different between the old and old+oil condition, the authors could not write that oils prevent in part the decrease in the MHC expressions.

It should be interesting to test the metabolic markers of figure 2 in the soleus muscle which is an oxidative muscle.

Is it a specific effect of oils in glycolytic muscle or on skeletal muscles with different metabolic and contractile phenotype? (gastro vs soleus).

Line 210, Myogenein replaced by myogenin

Discussion

In the first sentence of the discussion, the authors should be more careful in their suggestions, and then they should replace “prevent” by : “reduce the age-induced  impairments in skeletal muscle  physiology”.

This remark is true for all the discussion: example: “Oil mixture administration  prevented  age-induced  decrease  in  gastrocnemius  mass,  the  total  protein concentration and the gene expression of MHC”, this is not exactly showed in their results.

The paragraph concerning IGF-1 should be reduced in view of the results.

The link between HDAC-4 and myogenin has to be strengthen in the discussion based on the literature.

The central role played by HDAC-4 in the beneficial effect of mixt oil has to be reinforce based on the literature, and perhaps others central actors have to be more clearly discuss.

Line 378: What is exactly a “lower inflammatory status”? they have to use the term pro-and anti-inflammatory balance per example, or pro-inflammatory status or anti-inflammatory status.

Limit of the study must be developped.

Author Response

Remarks:

The dose of the oil mixture applied is rather high. What was the rationale for using such a dose?

The dose was chosen following previous studies of the group, in which we used similar protocols (Priego T. et al. 2013, Castillero et al. 2011). Nevertheless, other authors, with similar experimental designs, used also comparable doses (Kamolrat et al. 2013). Although the dose is high, the treatment consists in edible oils, suitable for consumption with no adverse effects.

  • Priego T,  Sánchez  J,  García  AP ,Palou  A,  Picó  Maternal  dietary  fat affects  milk  fatty  acid  profile  and impacts  on  weight  gain  and  thermogenic  capacity  of suckling rats. Lipids 2013; 48:481-495.
  • Castillero, E.; Lopez-Menduiña, M.; Martin, A.I.; Villanua, M.A.; Lopez-Calderon, A. Comparison of the effects of the n-3 polyunsaturated fatty acid eicosapentaenoic and fenofibrate on the inhibitory effect of arthritis on IGF1. J Endocrinol 2011, 210, 361-368, doi:10.1530/JOE-11-0170.
  • Kamolrat, T.; Gray, S.R.; Thivierge, M.C. Fish oil positively regulates anabolic signalling alongside an increase in whole-body gluconeogenesis in ageing skeletal muscle. Eur J Nutr 2013, 52, 647-657, doi:10.1007/s00394-012-0368-7.

Was the movement activity of the animals monitored?

No, we did not monitor the activity of the animals, but we agree with the reviewer that it would have be very informative.

Fig. 1C: mg/mg BW or per mg muscle weight?

We thank the reviewer for the observation, it was a mistake, it is “mg/100 mg muscle weight”. We have changed the figure accordingly (see Figure 1 in page 5).

Reviewer 2 Report

The manuscript presents data demonstrating inhibition of age-related changes in the

 gastrocnemius muscle and serum indices of inflammation by a 21-day treatment with a mixture of  algae oil (25%) and extra virgin olive oil. The experiments are generally well done and properly described. The results are interesting although they did not give a clue with respect to the active components of the oils which can be responsible for the effects observed.

Remarks:

The dose of the oil mixture applied is rather high. What was the rationale for using such a dose?

Was the movement activity of the animals monitored?

Fig. 1C: mg/mg BW or per mg muscle weight?

Author Response

In my opinion, this study is valid and suitable for publication. However, minor revisions/comments are strongly required:

  1. Why authors treated animals for 3 weeks? Is this treatment period sufficient to observe significant results? Is this reported in literature? Please, justify this experimental condition

Yes, we have previously described that this treatment administered during the same period of time improved the lipid profile, insulin sensitivity and vascular function in aged rats, and reduced endothelial dysfunction and vascular insulin resistance. These effects were associated to decreased inflammation and oxidative stress in the liver, and in the cardiovascular system. (González-Hedström et. al, 2020). Moreover, we have also reported that treatment with oils during a shorter period of time (12 days) is sufficient to observe beneficial effects on muscle (Castillero et al. 2009). Anyway, we believe that longer treatment periods may show greater beneficial responses.

  • Gonzalez-Hedstrom, D.; Amor, S.; de la Fuente-Fernandez, M.; Tejera-Munoz, A.; Priego, T.; Martin, A.I.; Lopez-Calderon, A.; Inarejos-Garcia, A.M.; Garcia-Villalon, A.L.; Granado, M. A Mixture of Algae and Extra Virgin Olive Oils Attenuates the Cardiometabolic Alterations Associated with Aging in Male Wistar Rats. Antioxidants (Basel) 2020, 9, doi:10.3390/antiox9060483.
  • Castillero E, Martín AI, López-Menduiña M, Villanúa MA & López-Calderón A 2009. Eicosapentaenoic acid attenuates arthritis-induced muscle wasting acting on atrogin-1 and on myogenic regulatory factors. American Journal of Physiology. Regulatory, Integrative and Comparative Physiology 297 R1322–R1331 doi:10.1152/ajpregu.00388.2009.
  1. Information about the treatment composition is needed. More specifically, EVOO does not include only PUFA, but also other bioactive compounds, including polyphenols, which anti-inflammatory and anti-ageing potential have been widely reported. Please, provide more details on chemical composition of the treatment used

In a previous work, we analyzed the composition of the oils used in the present paper (both, the composition of fatty acids and the phenolic fraction rich in polyphenols). This study has been recently published in the NFS Journal (“Protective effects of extra virgin olive oil against storage-induced omega 3 fatty acid oxidation of algae oil”, reference # 29 of the present article). In the revised version, we have added a comment about this matter in the materials and methods section (page 3, lines 110-113).

  • Gonzalez-Hedstrom, D.; Granado, M.; Inarejos-Garcia, A.M. Protective effects of extra virgin olive oil against storage-induced omega 3 fatty acid oxidation of algae oil. NFS Journal 2020, 21, 9-15, doi:10.1016/j.nfs.2020.08.003.
  1. Authors conclude that the anti-sarcopenia effects observed are mainly due to the anti-inflammatory activity of PUFAs. If so, what is the added value provided by this oil mixture? Probably, EVOO polyphenols may contribute to this aim. Please, see and eventually quote the following manuscript reporting the anti-sarcopenia effects of polyphenols in old rats: "Grape polyphenols ameliorate muscle decline reducing oxidative stress and oxidative damage in aged rats. Nutrients, 2020; 12(5):1280. doi: 10.3390/nu12051280"

We agree with the reviewer about the possible role of EVOO polyphenols on the beneficial effects observed. The suggested reference is of great interest and we have discussed these results in comparison with our results in the discussion section (see page 10, lines 298-301, reference # 39). We thank the reviewer for this interesting contribution.

  1. English language needs to be improved

We have improved the English language with the help of a professional native English speaker.

  1. Overall, the paper template needs to be respected in accordance with the journal authors guidelines (i.e. figure should be included in the text close the first citation, subsections should be entitled, etc...)

We have improved the template following the guidelines.

  1. Figure legends: please remove the double # if not reported in graph bars

We have done the changes suggested by the reviewer.

Reviewer 3 Report

This is an interesting study reporting the effect of oil mixture in the prevention of sarcopenia. Biochemistry and molecular biology experiments herein reported provide a solid base for the identification of putative MoA.

In my opinion, this study is valid and suitable for publication. However, minor revisions/comments are strongly required:

  1. Why authors treated animals for 3 weeks? Is this treatment period sufficient to observe significant results? Is this reported in literature? Please, justify this experimental condition
  2. Information about the treatment composition is needed. More specifically, EVOO does not include only PUFA, but also other bioactive compounds, including polyphenols, which anti-inflammatory and anti-ageing potential have been widely reported. Please, provide more details on chemical composition of the treatment used
  3. Authors conclude that the anti-sarcopenia effects observed are mainly due to the anti-inflammatory activity of PUFAs. If so, what is the added value provided by this oil mixture? Probably, EVOO polyphenols may contribute to this aim. Please, see and eventually quote the following manuscript reporting the anti-sarcopenia effects of polyphenols in old rats: "Grape polyphenols ameliorate muscle decline reducing oxidative stress and oxidative damage in aged rats. Nutrients, 2020; 12(5):1280. doi: 10.3390/nu12051280"
  4. English language needs to be improved
  5. Overall, the paper template needs to be respected in accordance with the journal authors guidelines (i.e. figure should be included in the text close the first citation, subsections should be entitled, etc...)
  6. Figure legends: please remove the double # if not reported in graph bars

Author Response

Major issues

In this article the scientific question is valid and the sample size is correct. The methods used by the authors are appropriate in order to answer the question.

In this article, sarcopenia is defined by “progressive decline in skeletal muscle mass, strength and function”, but there is no test to evaluate the loss of strength, so how can the authors be sure that animals developed sarcopenia? Loss of muscle mass is not systematically associated with the loss of strength.

We agree with the reviewer that not always loss of muscle mass is systematically associated with loss of strength. However, to our knowledge in aging induced-sarcopenia it is generally accepted that muscle atrophy is sarcopenia. Nevertheless, this can be one limitation of the study and accordingly we have included this issue in the discussion (see page 12, lines 400-402).

They did not mention the limitations of their study, it could be wise to mention it. Is there any adverse effects of the mixture oil for the rats?.

We did not observe any adverse effect of the oil mixture. Furthermore, the mixture oil administration ameliorate many alterations associated with aging such as: lipid profile, insulin resistance, endothelial dysfunction, as well as aging-induced inflammation and oxidative stress in both the liver and the cardiovascular system (ref # 31 of the present manuscript, González-Hedström et al. 2020).

  • Gonzalez-Hedstrom, D.; Amor, S.; de la Fuente-Fernandez, M.; Tejera-Munoz, A.; Priego, T.; Martin, A.I.; Lopez-Calderon, A.; Inarejos-Garcia, A.M.; Garcia-Villalon, A.L.; Granado, M. A Mixture of Algae and Extra Virgin Olive Oils Attenuates the Cardiometabolic Alterations Associated with Aging in Male Wistar Rats. Antioxidants (Basel) 2020, 9, doi:10.3390/antiox9060483.

Globally the language is clear and understandable, and the references are sufficient.

As mentioned in the minor issues, it should be important to precise the part of the gastrocnemius that the authors used. Moreover, it should be interesting to measure the same parameters in a second skeletal muscle with different muscle phenotype (ex. Soleus muscle).

We did not include the soleus muscle, because it has been reported that type I muscle is less affected by aging induced muscle wasting than type II muscles (Miljkovic et al. 2015).

  • Miljkovic N, Lim JY, Miljkovic I, Frontera WR. Aging of skeletal muscle fibers. Ann Rehabil Med. 2015 Apr;39(2):155-62. doi: 10.5535/arm.2015.39.2.155. Epub 2015 Apr 24. PMID: 25932410; PMCID: PMC4414960.

 The discussion has to be improved following several minor issues.

Minor issues

Introduction

The importance and the effects of IGF-1 impact on myogenesis and sarcopenia could be more developed, for a better understanding of IGF-1 part in the discussion. The scientific justification of algae oil utilization is poorly describing in the introduction.

We have added more information about the impact of IGF-I on myogenesis and sarcopenia in the introduction section (see page 2, lines 43-55). We have added a phrase in the introduction (page 2 and 3, lines 88-91) indicating the benefits of algae oil utilization.

Experimental materials

The authors can advise the number of animals used for each measure in the figure legend, and in the materials and methods. The authors written “Old male wistar (n=13)” then “half of the old rats”, so they have to precise the exact number.

We have specified the number of animals of each experimental group (see page 3, lines 106-113) and added that information in each of the figure legends.

Which part of the gastrocnemius was used? Because le superficial gastrocnemius is glycolytic and the deep part is more oxidative. The type of fiber is also different in function of the part of this muscle.

Gastrocnemius muscles were dissected and cut in sections. These sections contained both parts the red and white, as it is shown in the figure below, corresponding to a previous work we have done in inflammatory cachexia.

Representative crosssections of the midbelly region of the gastrocnemius, slow fibers are shown in red. (from Castillero et al 2011)

  • Castillero E, Nieto-Bona MP, Fernández-Galaz C, Martín AI,  López-Menduiña, Miriam Granado M,  Villanúa MA, and López-Calderón A. Fenofibrate, a PPARα agonist, decreases atrogenes and myostatin expression and improves arthritis-induced skeletal muscle atrophy. American Journal of Physiology-Endocrinology and Metabolism 2011 300:5, E790-E799.

Why the authors did not use several muscles in order to validate the decrease of muscle mass in some other hind limb muscles?

As many other authors that study skeletal muscle cachexia and/or sarcopenia, we have been several years studying muscle wasting in the gastrocnemius muscle, because it is a mixed muscle, that contains red (predominantly type IIa), and white (predominantly type IIb) fibers.  In addition, mixed muscles, (rich in type IIa fibers) seems to be more susceptible to oxidative stress (Voces et al. 2004).

  • Voces J, Cabral de Oliveira A.C., Prieto J.G., Vila L., Perez A.C., Duarte I.D.G. and Alvarez A.I. Ginseng administration protects skeletal muscle from oxidative stress induced by acute exercise in rats. 2004, Braz J Med Biol Res. 37:1863-71.

Experimental results

It could be relevant to give more precisions in the figure legend, for example with an informative title that resume the major result of the figure. They should mentioned the samples numbers for each graphs.

According to the referee suggestion, we have added a title in each figure that resume the major results of the data showed in the figure and, as mentioned above, we have added in the figure legend the number of samples used in each experimental group.

The authors should measure all the MHC in gastrocnemius and should give the results for the major MHC present in Gastrocnemius: the MHC-IIb/x.

We have analyzed two isoforms of the MHC present in gastrocnemius muscle and they show similar pattern of response to that of the gastrocnemius mass, we expect the same pattern in other MHC isoforms.

Because the results are not different between the old and old+oil condition, the authors could not write that oils prevent in part the decrease in the MHC expressions.

We agree with the reviewer and we have change the sentence accordingly (see page 4, lines 182-184).

It should be interesting to test the metabolic markers of figure 2 in the soleus muscle which is an oxidative muscle.

Is it a specific effect of oils in glycolytic muscle or on skeletal muscles with different metabolic and contractile phenotype? (gastro vs soleus).

As explained above, we did not include the soleus muscle, because it has been reported that type I muscle is less affected by aging.

Line 210, Myogenein replaced by myogenin

We thank the reviewer for this observation; we have change the word in the text.

Discussion

In the first sentence of the discussion, the authors should be more careful in their suggestions, and then they should replace “prevent” by : “reduce the age-induced  impairments in skeletal muscle  physiology”.

This remark is true for all the discussion: example: “Oil mixture administration  prevented  age-induced  decrease  in  gastrocnemius  mass,  the  total  protein concentration and the gene expression of MHC”, this is not exactly showed in their results.

As suggested by the reviewer, we have changed some sentences in the text accordingly (see page 9, line 272, line 274 and page 12, line 396).

The paragraph concerning IGF-1 should be reduced in view of the results.

It has been reduced and now it is shorter.

The link between HDAC-4 and myogenin has to be strengthen in the discussion based on the literature.

The central role played by HDAC-4 in the beneficial effect of mixt oil has to be reinforce based on the literature, and perhaps others central actors have to be more clearly discuss.

 As recommended by the reviewer, we have added new references in the discussion that point out the importance of the HDAC-4-myogenin axis in the muscle atrophy (see page 11, lines 353-357, references # 53,54). However, since the role of the oils on HDAC-4 is novel, there is a lack of information in the literature about this effect and, for this reason, we are not able to reinforce this part based on literature. However, the novelty of this study has been added to the discussion section (see page 11, lines 384-385).

Line 378: What is exactly a “lower inflammatory status”? they have to use the term pro-and anti-inflammatory balance per example, or pro-inflammatory status or anti-inflammatory status.

We have changed the sentence accordingly (see page 12, line 397).

Limit of the study must be developped.

We have added a final sentence in the discussion section about the limits of the study (see page 12, lines 400-402).